# Acetabular Peri-Prosthetic Fractures—A Narrative Review

**DOI:** 10.3390/medicina58050630

**Published:** 2022-05-01

**Authors:** Gautier Beckers, Az-Eddine Djebara, Morgan Gauthier, Anne Lubbeke, Axel Gamulin, Matthieu Zingg, Johannes Dominik Bastian, Didier Hannouche

**Affiliations:** 1Department of Orthopaedic Surgery and Traumatology, Geneva University Hospitals, 1205 Geneva, Switzerland; az-eddine.djebara@hcuge.ch (A.-E.D.); morgan.gauthier@hcuge.ch (M.G.); anne.lubbekewolff@hcuge.ch (A.L.); axel.gamulin@hcuge.ch (A.G.); matthieu.zingg@hcuge.ch (M.Z.); didier.hannouche@hcuge.ch (D.H.); 2Department of Orthopaedic Surgery and Traumatology, Inselspital, University Hospital of Bern, 3010 Bern, Switzerland; johannes.bastian@insel.ch

**Keywords:** acetabulum, fracture, peri-prosthetic, revision surgery, total hip arthroplasty

## Abstract

Acetabular peri-prosthetic fractures are rare but their incidence is rising due to the increased prevalence of total hip arthroplasty, the increasing life expectancy and the growing functional demand of an ageing population, the incidence of primary total hip arthroplasty is increasing. They are either intra-operative or post-operative and have various aetiologies. Several factors such as implant stability, bone loss, remaining bone stock, fracture pattern, timing, age and co-morbidities of the patients must be considered for adequate treatment. To date, the literature on this subject has been sparse and no universally recognized treatment algorithm exists. Their rarity makes them a little-known entity and their surgical management represents a challenge for most orthopaedic surgeons. This review aims to present an update on epidemiology, the diagnostic work up, existing classification systems, surgical approaches and therapeutic options for acetabular peri-prosthetic fractures.

## 1. Introduction

Unless managed properly, peri-prosthetic fractures of the acetabulum can have disastrous consequences for patients. Few reports assessed the incidence of these fractures as they are rare events. It is estimated that they occur in 0.07% of patients bearing a total hip arthroplasty (THA) [1]. Therefore, surgeons generally have little exposure to these fractures that are difficult to treat. Unlike acetabular peri-prosthetic fractures, femoral peri-prosthetic fractures are much more common with an incidence of 3.5% 10 years after index surgery [2].

Due to the aging of the population, an increasing number of THAs are performed each year. As a result, it is expected that the frequency of peri-prosthetic fractures, including those around the acetabulum, will increase in the coming years. It is estimated that by 2030, the annual number of primary THA procedures will grow by 173% in the United States (US) [3], and similar estimations have been made in Australia [4] and the United Kingdom (UK) [5].

Published articles addressing acetabular peri-prosthetic fractures consist mainly of small case series and case reports. These fractures require the surgeon to master high-end osteosynthesis and revision arthroplasty skills, which explains the difficulty experienced by many orthopaedic surgeons in their surgical management. The variability of implants used and the different infrastructural setups in hospitals further contribute to the lack of consensus in the treatment algorithms.

Peri-prosthetic acetabular fractures are mainly differentiated into two subtypes: intra-operative and postoperative.

*Intra-operative fractures* can occur during acetabular exposure, reaming, hip dislocation, and acetabular implant insertion or removal [6]. Their reported incidence varies between 0.09–0.4% [6,7]. Underreaming, blunt reamers, elliptical cups, porous tantalum cups, and previous irradiation of the pelvis are known risk factors for intra-operative fractures [6,7]. Additionally, the use of monobloc cups and lack of use of peri-operative fluoroscopy increase the probability of non-recognized intra-operative fractures.

*Post-operative fractures* may be due to traumatic events [1], infection [8], adverse local tissue reaction (ALTR) or debris-induced pelvic osteolysis [9] and pathological processes around the socket (such as neoplasm) [10,11]. In 0.9% of the cases, pelvic discontinuity (PD) is observed [12]. Osteoporosis is an accepted risk factor for post-operative fractures [13] while age is still debated [13,14].

The objective of this review is to share our experience and to propose an algorithm for the management of periprosthetic acetabular fractures based on recently published articles.

## 2. Classification of Peri-Prosthetic Acetabular Fractures

Several classifications have been proposed over the past years. In 1996, Peterson and Lewallen [1] were the first to classify peri-prosthetic acetabular fractures into two groups: Type 1, those with a clinically and radiologically stable acetabular component and type 2, those with an unstable acetabular component.

In 1998, based on in vitro investigations, intraoperative peri-prosthetic acetabular fractures were classified by Callaghan et al. [15] into four groups: type A for anterior wall, type B for transverse, type C for inferior lip and type D for posterior wall fractures.

The most commonly used classification to date is the modified 2003 Paprosky and Della Valle classification for peri-prosthetic acetabular fractures (Figure 1) [16]. It is a comprehensive classification system incorporating all known variants (intra-operative during component insertion, intra-operative during component removal, traumatic fractures, spontaneous fractures, pelvic discontinuity) and providing guidelines on surgical and non-surgical fracture management. This classification is distinct from the Paprosky classification of acetabular defects.

A novel classification introduced by Pascarella et al. [17] in 2018 takes into account the timing of the fracture: intraoperative or postoperative, as well as the implant stability. Its main advantage is its ease of use and associated treatment algorithm. This classification has not been used by other authors.

Another classification worth mentioning is the unified classification system (UCS) that was introduced in 2014 and is used by the AO. It considers the stability of the implant, the fracture location and its anatomical features [18]. However, because of its complexity and lack of therapeutic implication, its clinical use is limited.

The Judet and Letournel classification is the most widely used classification system for acetabular fractures. Acetabular fractures are divided into 10 patterns: 5 simple patterns (posterior wall, posterior column, anterior wall, anterior column and transverse) and 5 complex patterns (posterior column with posterior wall, transverse posterior wall, T-style, anterior column posterior hemi-transverse and both columns) [19].

## 3. Diagnosis

The patient’s medical history, as well as previous radiographs, should be obtained, whenever possible. Pre-existing pain could indicate septic or aseptic loosening that could alter the therapeutic approach.

Imaging should include antero-posterior plain pelvic radiographs in all cases. In case of suspected and/or recognized fracture line, inlet, outlet and Judet recommend adding radiographs (internal and external 45-degree rotation of the pelvis) to further assess the fracture pattern. They should be compared to pre-traumatic radiographs when available.

For post-operative fractures, a computed tomography scan with metal suppression and three-dimensional reconstruction is mandatory to look for radiologic signs of loosening and infection (osteolysis, periosteal reaction, bone resorption around the implant), and to evaluate the fracture pattern, the bone stock, the position, and the stability of the cup.

Pre-operative vascular assessment by angio-CT is mandatory for all patients with intra-pelvic cup migration and/or clinical suspicion of arteriovenous injury to diagnose potential vascular lesions [20]. Migration of an implant beyond the ilioinguinal line may be systematically associated with a displacement of the external iliac vessels [21].

If peri-prosthetic THA infection is suspected, joint aspiration and analysis of the synovial fluid should be performed for cell count and microbiological culture [22].

## 4. Treatment

Patient- and fracture-related factors must be considered when choosing the best treatment for each patient. Factors include age, comorbidities, timing, implant stability, bone loss and fracture pattern, among others. The appropriate treatment will vary depending on the personality of the patient and the personality of the fracture.

For this review, we will focus on the treatment of Paprosky and Della Valle type I, III and V fractures which are the most frequent. Because of their extreme rarity, Paprosky and Della Valle type II and IV fractures will not be discussed.

### 4.1. Intra-Operative Fractures

Intra-operative fractures might be difficult to detect. They should be suspected when the acetabular component does not achieve the expected press-fit fixation, or when the acetabular component protrudes more medially upon impaction. The treatment depends on the displacement of the fracture and the stability of the acetabular component.

*Undisplaced intra-operative fractures* with stable acetabular component (Pascarella type 1a and Paprosky and Della Valle IA) can be treated conservatively by toe-touch weight or non-weight bearing for 6–10 weeks (Figure 2) [7,10,17,23]. If recognized during the surgery, the use of supplemental screws is recommended [23]. In those cases, a multi-hole cup is used as a plate and the component is stabilized by several posterosuperior (ilium) and posteroinferior (ischium) bicortical screws [7,17]. Some authors suggest using autografts from the iliac crest to speed up the healing process [7]. For some authors, non-recognized intraoperative fractures of the acetabulum (Paprosky and Della Valle type IC) have the same outcome that noncomplicated THA without further treatment [24]. For others, non-recognized intraoperative fracture of the acetabulum leads to loosening and cup migration in 100% of the cases [7].

*Displaced and/or unstable Intraoperative recognized cups* (Pascarella type 1b, Paprosky and Della Valle type IB) confront the surgeon with a tough decision. First, the component should be removed, the fracture line should be highlighted, and intra-operative radiographs should be obtained. Second, the integrity of both anterior and posterior columns should be assessed. If there is an important motion at the fracture site, the use of a cup as a plate is not effective and fixation of the fracture with standard reconstruction plates should be done prior to acetabular component insertion. One of the difficulties in the management of intra-operative fractures is to evaluate if fracture fixation can be performed adequately through the same approach. In most cases, the posterior wall is involved and might require fixation. If THA is performed using a direct anterior approach, an additional surgical approach might be needed to carry out the surgery. In our experience, transverse, posterior wall and posterior column fractures as well as femoral and acetabular component changes can be addressed through a posterior approach such as the Kocher–Langenbeck. For anterior column fractures, the ilio-inguinal, ilio-femoral, modified Stoppa or Pararectus approach might be necessary [25]. The ilio-inguinal approach remains the gold standard to address anterior acetabulum fractures. When performed correctly, it provides good exposure and is relatively atraumatic [25]. However, the modified Stoppa approach and the Pararectus approach are the authors’ preferred approaches to manage the anterior acetabulum, as they provide a good exposure of the pelvic ring [26,27,28].

### 4.2. Post-Operative Fractures

Post-operative fractures (Pascarella type 2, Paprosky and Della Valle type III) are divided into groups based on the stability of the cup. They are either acute, after a traumatic event, or chronic, due to osteolytic bone loss.

*Non-displaced stable fractures* (Pascarella type 2a, Paprosky and Della Valle type IIIA) can be treated non-operatively with 6–8 weeks of non-weight bearing [17]. Despite the paucity of articles treating this subject, conservative treatment carries a poor prognosis (Figure 3). In an article published in 1996, Peterson and Lewallen [1] reported that six out of eight patients treated non-operatively required surgery at 20 weeks, two for non-union and four for loosening despite fracture healing. Some authors cite open reduction and internal fixation (ORIF) as an appropriate treatment option for non-displaced acute fractures [17].

*In displaced fracture with unstable cups* (Pascarella type 2b, Paprosky and Della Valle type IIIB), surgery is needed [29]. The type of fracture according to the classification of Letournel [19] dictates the strategy. It is recommended to fix the fracture with standard reconstruction plates followed by a revision shell with multiple screws. Bone grafting might be necessary to fill the gaps at the fracture site (Figure 4) [17,30,31]. Several methods can be used to fix the fracture depending on the surgeon’s preference, including posterior column plating, anterior and posterior column plating, anterograde or retrograde anterior or posterior column screw fixation. Alternatively, in cases where primary fixation of the revision shell cannot be achieved, or if plating provides insufficient stability, Kerboull reinforcement rings, cup-cage constructs or antiprotusio cages can be used with cemented dual-mobility cups, as described later (Figure 5).

*In chronic and osteolytic contexts* (Pascarella type 2c), prosthetic revision with bone loss restoration is advised [17]. For this situation, we recommend the use of cup-cage constructs or antiprotrusio cages. Bone stock loss needs to be considered in these fractures and should be treated with allograft impaction grafting in contained defects, or with either bulk allografts or metallic augments when the acetabular rim is involved. In major osteolytic defects, and when all other options cannot be envisaged, stemmed acetabular cups anchoring into the iliac isthmus have been taken into consideration as a salvage solution. However, this type of implant which is used mainly in tumour resection surgery should be considered carefully, as high complication rates have been reported at five years, 31% of which require re-intervention [32,33].

### 4.3. Pelvic Discontinuity (PD)

Pelvic discontinuity (Paprosky and Della Valle type V) is a “distinct form of bone loss, occurring in association with THA, in which the superior aspect of the pelvis is separated from the inferior aspect because of bone loss or a fracture through the acetabulum” [12]. The preoperative diagnosis of this rare entity, found in 0.9–2.1% of acetabular revisions is, in the majority of the cases, not obvious [12,34,35].

PD can be distinguished into two entities, acute and chronic, each with a different potential for healing and biological bone ingrowth, thus requiring a different therapeutic approach. Furthermore, the percentage of bone loss needs to be incorporated into the treatment algorithm for PD. According to the Letournel classification, acute PD can be observed in both column, transverse, T-type, anterior column + posterior hemitransverse, and transverse + posterior wall patterns. Acute PD is either traumatic, typically occurring after a fall, or iatrogenic, following overreaming, implant impaction and removal [36]. The management of acute PD follows the same principles as those applied for postoperative fractures, and should be guided by the pattern of the fracture. In transverse fractures, ORIF of the posterior column and the use of a hemispherical acetabular shell can be successfully used to achieve good stability [34]. In the acute setting, the posterior plate provides extra-acetabular compression. Rogers and al. [34] reported 100% survivorship (no revision surgery, no reported infections and/or dislocations) on nine patients with acute PD with a mean follow-up of 34 months with this technique. Although not specifying if used in chronic or acute PD, Martin et al. [37] reported 80% revision-free survivorship of the posterior column plating technique with 22% complications, mostly infection and dislocation. In fractures of the anterior column, or in very unstable fractures such as in both columns and T-type acetabular fractures, an additional fixation with a pelvic reconstruction plate can be necessary to achieve rigid fixation of the fracture [38] (Figure 5).

Chronic PD is progressive and occurs because of periprosthetic septic/aseptic osteolysis or age-related osteopenia [36]. Despite numerous publications and multiple treatment options, no universally recognized treatment algorithm exists. Their management differs significantly from that of acute peri-prosthetic fractures as successful reconstruction will depend on the amount of the remaining bone stock, the ability to achieve a stable cup fixation, and the healing potential of the discontinuity.

In chronic PD, the pelvis is very stiff, and specific reconstruction techniques might be required to achieve implant stability. Chronic PD with moderate bone loss (Paprosky and Della Valle type VA) and good bone quality can be treated with a posterior compression plate, bone grafting, and a revision shell acting as a plate. Alternatively, a Kerboull reinforcement device and bulk allograft can be used together with a cemented cup. With re-revision surgery as an endpoint, an excellent outcome with a 15-year survival rate of 85% has been reported [39].

In the treatment of PD with severe bone stock loss (Paprosky and Della Valle type VB and VC), the use of acetabular cages alone comes with unfavourable results and high failure rates up to 76% [40,41,42], due to mechanical loosening and fatigue fracture of the implant [40]. Therefore, the use of acetabular cages is not recommended in these situations [43]. Alternatively, three other techniques and devices could be considered:The “Cup-cage construct” technique, currently the most popular treatment of chronic PD [42], was first described by Hanssen and Lewallen in 2005 [44]. It consists of an ilio-ischial cage, placed over an uncemented highly porous metal cup. In a majority of the cases, “jumbo cups”, defined by von Roth et al. [45] as an acetabular component with an outside diameter ≥66 mm in men and ≥62 mm in women, are used and thus help restore the centre of rotation (COR) of the hip in an anatomic position [46,47]. Remaining bone defects can be filled with augments or allograft. The cage offers initial stability and allows the osteointegration of the acetabular component. A polyethylene liner is then cemented in the cage in the correct position. Advantages of this technique are its favourable outcomes and high survival rates, ranging from 75–100% [37,42,48,49,50,51,52]. The main complications of this technique are dislocation (7–8%), infection (4–7%) and aseptic loosening (4–15%) [37,42,48,49].Acetabular distraction was first described by Sporer et al. in 2012 [53]. The acetabulum is reamed until the antero-superior and postero-inferior margins are engaged. Remaining bone defects are filed with porous tantalum augments. An acetabular component of the same material, 6–8 mm larger than the last reamer is then impacted. The distraction creates a press fit and a pelvic recoil as a result of ligamentotaxis [54]. The latter in conjunction with multiple screws inserted in the remaining ilium and ischium provides initial stability. The polyethylene liner or a dual mobility cup is then cemented into the shell [53]. Although relatively new, acetabular distraction is a promising treatment for chronic PD. Excellent results [40,42,53,55], with low complication rates, 3–5% aseptic loosening [40,53], have been reported at 2- to 7-year follow-up.Custom-made triflange implants are another option to address chronic PD with severe bone loss. Based on a preoperative CT scan, an individually produced titanium, porous and/or hydroxyapatatite-coated triflange cup is made. Through the fixation of the three flanges (ilial, ischial and pubic), initial stability with the hip COR in anatomic position can be achieved. Excellent results and >80% survivorship of the implants are reported [42,56,57,58]. The disadvantages of this implant are high costs, long manufacture time (6 weeks) and the high rates of dislocation, up to 21% [42,56,57].

The strength of this article is its clinical application. The treatment algorithm provides, based on our experience and the current literature, an adequate therapeutic solution for the different fracture types.

To the best of our knowledge, no treatment algorithm for acetabular peri-prosthetic fractures has been published so far.

The weaknesses of the article are twofold. First, the proposed treatment algorithm has not yet been validated by a study. Second, the article is a narrative and not a systematic review with the ensuing biases.

The authors preferred treatment algorithm is summarized in the figures below (Figure 6, Figure 7 and Figure 8).

## 5. Conclusions

Acetabular peri-prosthetic fractures are a rare complication of THA with disastrous consequences for the patient if not managed properly. Their surgical management represents a challenge for most orthopedic surgeons. The treatment algorithm includes the assessment of the timing, implant stability, bone loss and fracture pattern and provides an individualized therapeutic strategy.

## Figures and Tables

**Figure 1 medicina-58-00630-f001:**
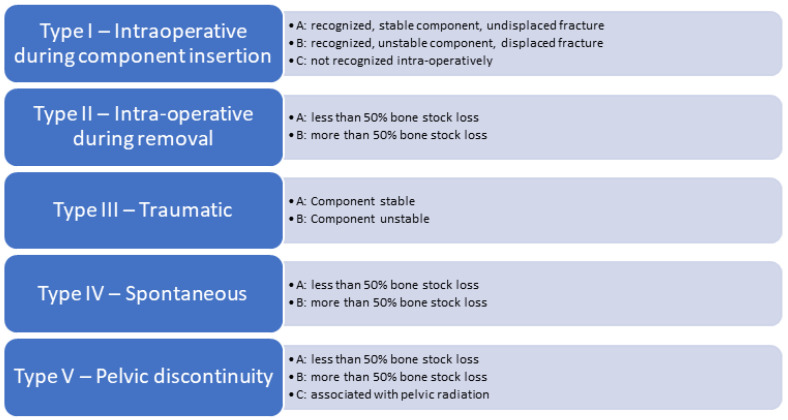
Modified 2003 Paprosky and Della Valle classification for periprosthetic acetabular fractures.

**Figure 2 medicina-58-00630-f002:**
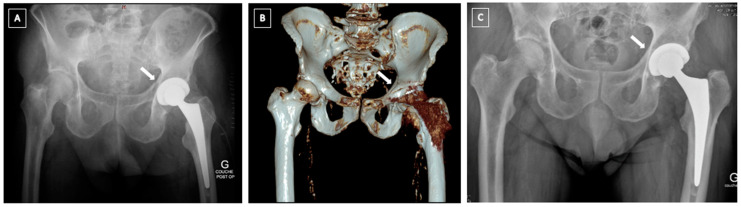
Intra-operative fracture of the left acetabulum recognized on immediate post-operative X-rays. It was treated non-operatively with 8 weeks of non-weight bearing with a favourable outcome. (**A**) postoperative CT scan showing the non-displaced fracture of the anterior column according to Judet and Letournel’s classification; (**B**) Immediate postoperative X-ray; (**C**) X-ray at 1 year follow-up.

**Figure 3 medicina-58-00630-f003:**
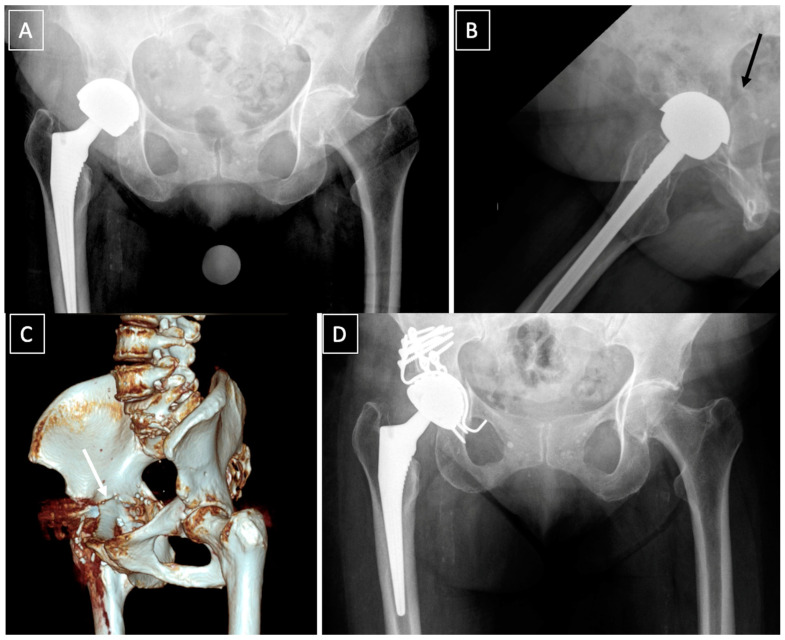
Failure of a conservative treatment of an acute post-operative fracture (black arrow) (**A**,**B**), following a low energy trauma, at 6 months. (**C**): 3D CT scan showing a nonunion of a transverse fracture of the acetabulum (white arrow). (**D**): postoperative X-ray showing a plating of the posterior column and reconstruction with a Kerboull acetabular device and a cemented dual-mobility cup. Reaming to activate nonunion was performed and bone graft added. Septic non-union was ruled out with intraoperative samples.

**Figure 4 medicina-58-00630-f004:**
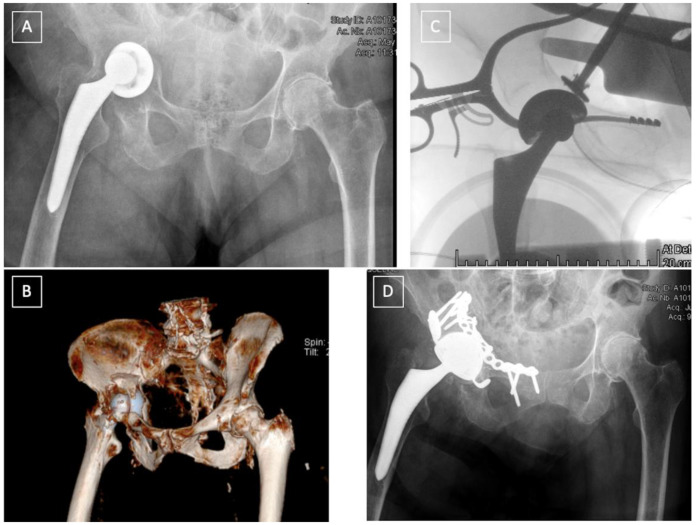
Acute post-operative fracture, following a low energy trauma, of the anterior column in a 94-year-old patient (**A**,**B**) treated by one stage surgery. Reduction and ORIF of the anterior column through a Stoppa approach, and revision of the cup through a direct anterior approach with reconstruction of the acetabulum and a cemented dual-mobility cup (**D**). Reduction of the cup can be achieved from the Stoppa approach (**C**).

**Figure 5 medicina-58-00630-f005:**
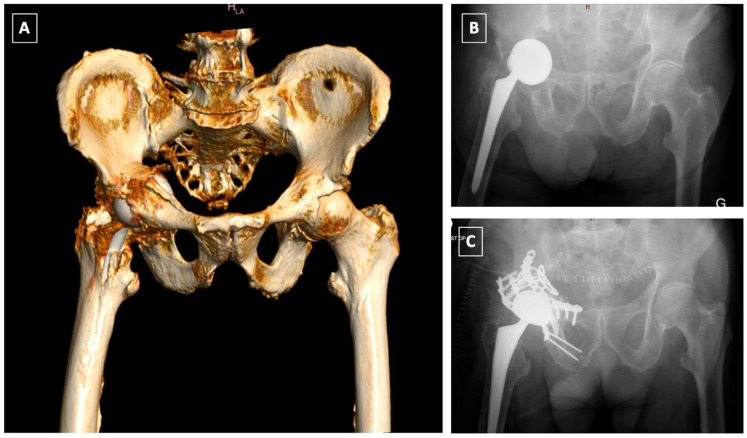
Late postoperative T-type fracture (**A**,**B**): ORIF of the anterior column using a Stoppa approach followed by ORIF of the posterior column and acetabular reconstruction using a Burch-Schneider Antiprotrusio cage via a Kocher-Langenbeck approach (**C**).

**Figure 6 medicina-58-00630-f006:**
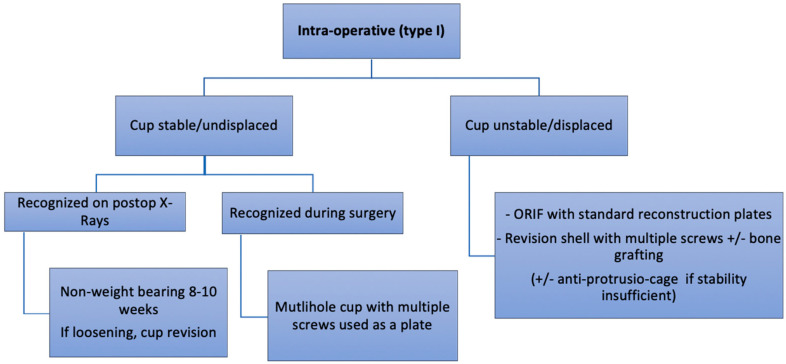
Authors preferred treatment algorithm for intra-operative fractures.

**Figure 7 medicina-58-00630-f007:**
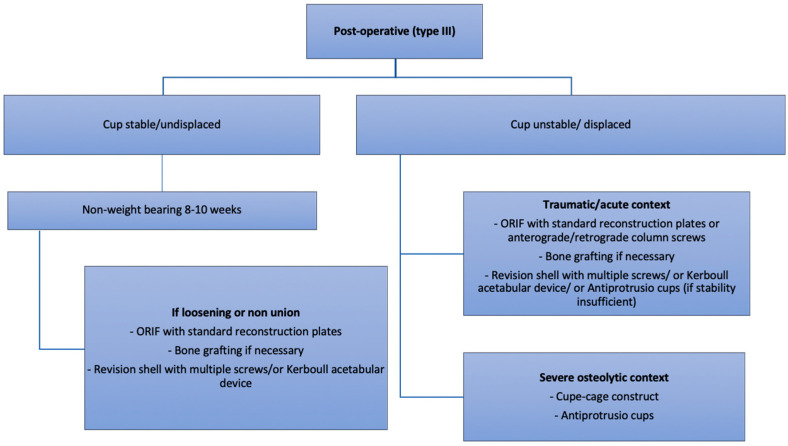
Authors preferred treatment algorithm for post-operative fractures.

**Figure 8 medicina-58-00630-f008:**
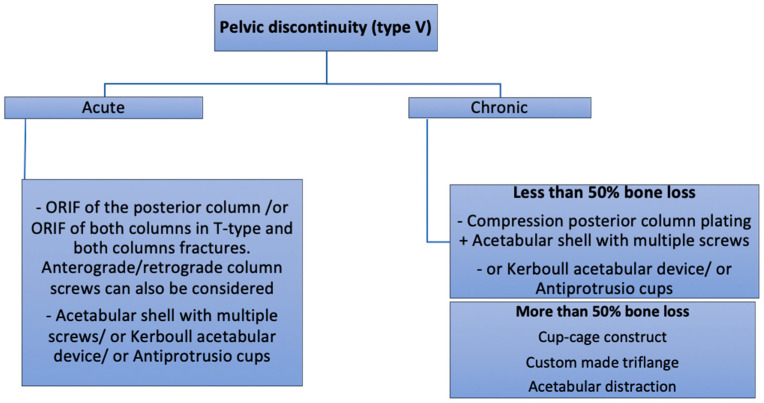
Authors preferred treatment algorithm for pelvic discontinuity.

## Data Availability

Not applicable.

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
