# Peer review of "Acetabular Peri-Prosthetic Fractures—A Narrative Review"

_medicina, 2022, doi:10.3390/medicina58050630_

Round 1

Reviewer 1 Report

Dear Authors, this narrative review is completed e very spread. your proposal of algorithm will be validated using a case series in future work

What is the main question addressed by the research? 
the Authors conducted a review of literature regarding different aspects of periprosthetic acetabular fractures.
Is it relevant and
interesting? 
the work is very interesting and complete
How original is the topic? 
the topic is original (4/5; 5 is maximum)
What does it add to the subject area compared with other published material? 
the Authors suggested a specific possibility of treatment according to algorithm
Is the paper well written?
the paper is well written
Is the text clear and easy to read? 
different sections are well balanced and well written
Are the conclusions consistent with the evidence and arguments presented? 
the description of different treatments is clear and the Authors suggested the good treatment by results of literature
Do they address the main question posed?
The Authors reached the main aim focalizing the attention on differents treatment

Author Response

What is the main question addressed by the research? 
the Authors conducted a review of literature regarding different aspects of periprosthetic acetabular fractures.
Is it relevant and interesting? 
the work is very interesting and complete

Answer: Thank you very much for your comments.
How original is the topic? 
the topic is original (4/5; 5 is maximum)

Answer: Thank you very much for your comments.
What does it add to the subject area compared with other published material? 
the Authors suggested a specific possibility of treatment according to algorithm

Answer: Thank you very much for your comments.
Is the paper well written?
the paper is well written

Answer: Thank you very much for your comments.
Is the text clear and easy to read? 
different sections are well balanced and well written

Answer: Thank you very much for your comments.
Are the conclusions consistent with the evidence and arguments presented? 
the description of different treatments is clear and the Authors suggested the good treatment by results of literature

Answer: Thank you very much for your comments.
Do they address the main question posed?
The Authors reached the main aim focalizing the attention on differents treatment

Answer: Thank you very much for your comments.

English language and style are fine/minor spell check required

Answer: Linguistic and spelling revisions were performed throughout the manuscript.

Reviewer 2 Report

The manuscript is a narrative review of periprosthetic acetabular fractures (intra- and post-operative), a situation that orthopaedic surgeons are increasingly confronted with due to the steady increase in hip replacement surgery.

The review is well structured and also provides a final treatment algorithm proposed by the authors. 
It only needs a minor linguistic revision. 

The following are some suggestions:

- On line 84, after "infection" if the authors consider it necessary you could mention this work Basilico M, Vitiello R, Oliva MS, Covino M, Greco T, Cianni L, Dughiero G, Ziranu A, Perisano C, Maccauro G. Predictable risk factors for infections in proximal femur fractures. J Biol Regul Homeost Agents. 2020 May-Jun;34(3 Suppl. 2):77-81. PMID: 32856444 and after "neoplasm", together with citation 9, could add this Vitiello R, Bocchi MB, Gessi M, Greco T, Cianni L, de Maio F, Pesce V, Maccauro G, Perisano C. Induced membrane by silver-coated knee megaprosthesis: keep or toss? J Biol Regul Homeost Agents. 2020 Sep-Oct;34(5 Suppl. 1):101-106. IORS Special Issue on Orthopedics. PMID: 33739013. 
- I recommend a minor linguistic and spelling revision, such as on line 140 where a dot is missing after the reference [18].

After these corrections, this review may conform to the standards of the journal and therefore be worthy of publication.

Author Response

The review is well structured and also provides a final treatment algorithm proposed by the authors. 
It only needs a minor linguistic revision. 

Answer: 

Thank you very much for your comments.

Linguistic and spelling revisions were performed throughout the manuscript.

The following are some suggestions:

  • On line 84, after "infection" if the authors consider it necessary you could mention this work Basilico M, Vitiello R, Oliva MS, Covino M, Greco T, Cianni L, Dughiero G, Ziranu A, Perisano C, Maccauro G. Predictable risk factors for infections in proximal femur fractures. J Biol Regul Homeost Agents. 2020 May-Jun;34(3 Suppl. 2):77-81. PMID: 32856444 and after "neoplasm", together with citation 9, could add this Vitiello R, Bocchi MB, Gessi M, Greco T, Cianni L, de Maio F, Pesce V, Maccauro G, Perisano C. Induced membrane by silver-coated knee megaprosthesis: keep or toss? J Biol Regul Homeost Agents. 2020 Sep-Oct;34(5 Suppl. 1):101-106. IORS Special Issue on Orthopedics. PMID: 33739013.  

    Answer: Thank you for pointing out these 2 references. They were added accordingly.

    - I recommend a minor linguistic and spelling revision, such as on line 140 where a dot is missing after the reference [18]. Answer: Linguistic and spelling revisions were performed throughout the manuscript.

Reviewer 3 Report

This is a relatively good study. However, a major revision is still needed. Some suggestions are given as belows:

  1. Results section should be more concise. The parameters of predicting surgical effect should be further analyzed, including the correlation

  2. The results, analysis and discussion should be combined with clinical effect of surgery.

  3. What makes your study special? What do you conclude on your study results?

  4. This study has many limitations. Mention the strengths and weaknesses of your study in detail.

  5. Discussion is improper.

  6. Add References below

PMID: 28869754

PMID: 27471388

PMID: 30644789

Author Response

Thank you very much for your comments.

Linguistic and spelling revisions were performed throughout the manuscript.

This is a relatively good study. However, a major revision is still needed. Some suggestions are given as belows:

  1. Results section should be more concise. The parameters of predicting surgical effect should be further analyzed, including the correlation. Answer: We do not have the impression that they apply to the type of article that we submitted.

  2. The results, analysis and discussion should be combined with clinical effect of surgery. Answer: We do not have the impression that they apply to the type of article that we submitted.

  3. What makes your study special? What do you conclude on your study results? Answer: We added a paragraph mentioning the strength and limitations of our article.

  4. This study has many limitations. Mention the strengths and weaknesses of your study in detail. Answer: We added a paragraph mentioning the strength and limitations of our article.

  5. Discussion is improper. Answer: We do not have the impression that they apply to the type of article that we submitted.

  6. Add References below

PMID: 28869754

PMID: 27471388

PMID: 30644789

Answer: Thank you for pointing out these references. In our opinion even if they are very interesting, they are to specific and not quite in line with the “Main” subject of the article.